# FOOL YOUR (VISION AND) LANGUAGE MODEL WITH EMBARRASSINGLY SIMPLE PERMUTATIONS

## ABSTRACT

Large language and vision-language models are rapidly being deployed in practice thanks to their impressive capabilities in instruction following, in-context learning, and so on. This raises an urgent need to carefully analyse their robustness so that stakeholders can understand if and when such models are trustworthy enough to be relied upon in any given application. In this paper, we highlight a specific vulnerability in popular models, namely permutation sensitivity in multiple-choice question answering (MCQA). Specifically, we show empirically that popular models are vulnerable to adversarial permutation in answer sets for multiple-choice prompting, which is surprising as models should ideally be as invariant to prompt permutation as humans are. These vulnerabilities persist across various model sizes, and exist in very recent language and vision-language models.

## 1 INTRODUCTION

Large language models (LLMs) (Brown et al., 2020; OpenAI, 2023a; Touvron et al., 2023a) and large vision-language models (VLLMs) (Alayrac et al., 2022; Li et al., 2023c) have made astonishing progress in recent years. They have attained strong capabilities across a diverse array of language tasks, enabling nuanced text generation, sophisticated instruction following, and natural dialogue with multimodal input and output. One task where they demonstrate particular prowess is multiple-choice question answering (MCQA) (Robinson & Wingate, 2023). This is an important capability with many real-world applications, from education to recruitment exams. Current LLMs and VLLMs have widely utilized the task format of MCQA for benchmarking and evaluation (Hendrycks et al., 2020; Lu et al., 2022; Zhong et al., 2023; Liang et al., 2022; Schwenk et al., 2022). This has built confidence that they can generate accurate and robust answers, underpinned claims of LLM competence at professional level human qualifications such as the bar exam (OpenAI, 2023b), and even led to reports of surpassing human-level performance on various tasks.

Surprisingly, contrary to the confidence instilled by high-performance metrics on established benchmarks, these models are surprisingly brittle when subjected to simple permutations in the answer choices, i.e., randomly changing the option positions. In this paper, we show that even a simple permutation of the answer sets, as illustrated in Figure 1, can lead to a dramatic decline in accuracy for both LLMs and VLLMs in a wide range of MCQA datasets, sometimes even below the random

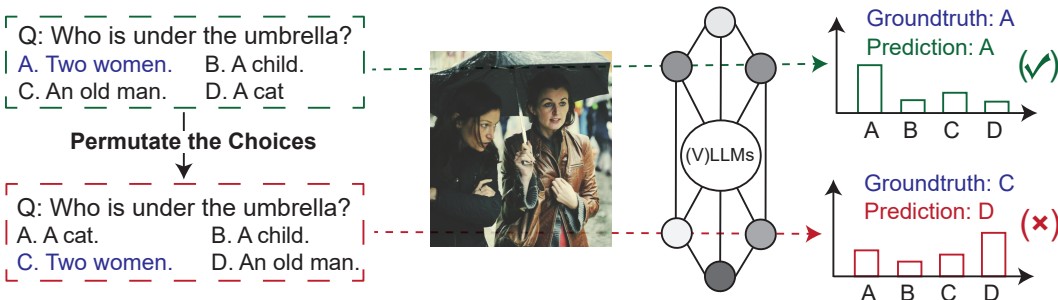

Figure 1: Schematic Illustration of an MCQA permutation attack.

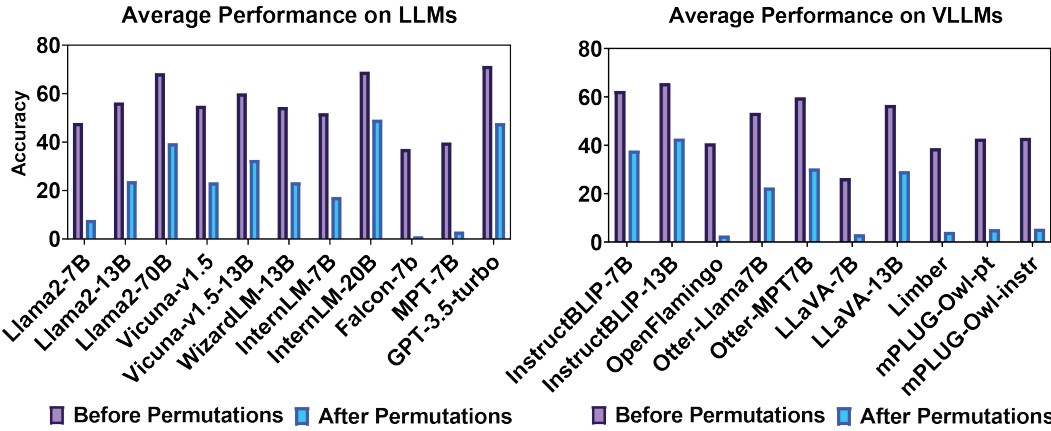

Figure 2: Summary of MCQA adversarial attack results for both LLMs and VLLMs. The values are average accuracy across all benchmarking datasets.

chance levels. For instance, Llama2-13B (Touvron et al., 2023a) experiences a 33.89% degradation in accuracy on the MMLU dataset (Hendrycks et al., 2020) following random permutation of option positions, with results falling below the random chance. A wide variety of popular LLMs and VLLMs, suffer significantly from this vulnerability, as summarised in Figure 2.

Furthermore, our investigations reveal an even more disconcerting aspect: the vulnerability to permutations persists in LLMs and VLLMs even when multiple distractor options are deliberately removed from the answer sets. Intuitively, one expects that by eliminating incorrect choices, the task should become simpler due to increasing chance performance, thereby enhancing the models' performance. However, our empirical findings contradict this notion. Even with a reduced number of distractors, the performance of both LLMs and VLLMs remains susceptible to degradation, affirming the deeply ingrained nature of this vulnerability.

To further investigate the source of the brittleness, we demonstrate through our adversarial attack that it is not merely a selection bias towards/against certain positions, such as moving correct answers to a fixed position that a given model is biased against picking. While positional factors may moderately influence model performance, they do not explain the strength of our adversarial attack results, suggesting a more systemic issue that extends beyond simple position bias.

This issue should be of intrinsic concern to those seeking to understand and design trustworthy and reliable LLMs and VLLMs, or emulate human capabilities. However, one might speculate that the issue could be mitigated in practice through the engineering solution of majority voting across different permutations or by employing calibration strategies as suggested in previous work (Zhao et al., 2021). However, our findings indicate that while majority voting may offer some degree of improvement, the resulting performance still lags behind the original metrics, despite incurring a $k! \times$ computational cost of the original inference time. Additionally, calibration techniques such as calibrate-before-use (Zhao et al., 2021) fail to alleviate this problem effectively.

In summary, our research unveils a glaring yet often overlooked vulnerability in large language models and vision-language models, specifically within the domain of multiple-choice question answering (MCQA). Despite their impressive metrics on well-established benchmarks, these models reveal a disconcerting fragility when faced with simple manipulations such as option permutations. Existing mitigation strategies fall short of effectively resolving this issue. Our observations not only raise pivotal questions about the models' robustness but also accentuate the necessity for heightened scrutiny in assessing their MCQA capabilities. We argue that stakeholders should be vigilant in relying on such models until these vulnerabilities are adequately addressed.

## 2    SIMPLE ADVERSARIAL ATTACK BREAKS LLMs AND VLLMs

In this section, we analyse the brittleness of a broad array of large language models and vision-language models to random adversarial attacks in MCQA. By simply shuffling answer choices, we find that these models fail to maintain their performance, revealing a critical vulnerability.

### 2.1    EXPERIMENT SETUP

In an ideal scenario, robust models should offer consistent predictions that are *invariant* to permutations that have no semantic influence on the question being posed. To test this, we simply iterate through the possible permutations of MCQ options. A robust model should be correct in every case. While there are $k!$ possible combinations in total, we cease permutation once the model produces an incorrect prediction (succumbs to the permutation attack), which usually requires far less than $k!$ attempts[1].

Formally, Given a question $q$ and an answer list $A = \{a_1, a_2, \ldots, a_k\}$, the permutation adversarial attack can be described by the Equation 1. We maximize the loss function ($\mathcal{L}$) with respect to all possible permutations ($\Pi$) of the answer list. Here, prompt$(q, A)$ prompts the model with the given query and answer list, and the model's response is then evaluated by the loss.

$$\text{Maximize:} \quad \mathcal{L}\left(\text{prompt}(q, A^*)\right)$$
$$\text{s.t.} \quad A^* \in \Pi(A) \tag{1}$$

Table 1: Statistics of the language datasets evaluated.

|  | **MMLU** | **ARC-c** | **BoolQ** | **SocialiQA** | **MedMCQ** |
|---|---|---|---|---|---|
| # of choices | 4 | 4 | 2 | 3 | 4 |
| # QA pairs | 14079 | 1165 | 3270 | 1954 | 2816 |
| Task | Aggregated | Commonsense Reasoning | Reading Comprehension | Commonsense Reasoning | Out-of-domain |

**Models**    We evaluate a wide range of LLMs and VLLMs of diverse sizes, different pretrained backbones, and both auto-regressive pretrained and instruction-following fine-tuned models. Specifically, for LLMs, we have evaluated LLaMA-2 (7B/13B) (Touvron et al., 2023b), Vicuna (7B/13B) (Chiang et al., 2023), WizardLM-13B (Xu et al., 2023), InternLM-20B (Team, 2023a), Falcon-7B Penedo et al. (2023), and MPT-7B (Team, 2023b). For VLLMs, InstructBLIP (Vicuna-based, 7B/13B) (Dai et al., 2023), Open-Flamingo (MPT-based, 9B) (Awadalla et al., 2023), Otter (Llama-based, MPT-based) (Li et al., 2023a), LLaVA

Table 2: Statistics of the vision-language datasets evaluated.

|  | # of choices | # QA pairs |
|---|---|---|
| **ScienceQA** | 2,3,4,5 | 2021 |
| **A-OKVQA** | 4 | 1145 |
| **MMBench** | 4 | 4377 |
| **SEED-Bench** | 4 | 14233 |

(7B/13B) (Liu et al., 2023a), Limber (7B) (Merullo et al., 2023), and mPLUG-Owl (pretraining, intruction) (Ye et al., 2023) are used for evaluation.

**Datasets**    We utilize a diverse array of language and vision-language MCQA datasets for comprehensive evaluation. These datasets cover multiple domains and require different aspects of the models to give correct answers, ensuring our findings are generalizable. Specifically, for LLMs, we utilize MMLU (Hendrycks et al., 2020), ARC challenge (ARC-c) (Clark et al., 2018), BoolQ (Clark et al., 2019), SocialiQA (Sap et al., 2019), and MedMCQA (Pal et al., 2022). For VLLMs, we use ScienceQA (Lu et al., 2022), A-OKVQA (Schwenk et al., 2022), MMBench (Liu et al., 2023c),

---

[1]Since typical MCQA benchmarks use $k = 4$, the brute force algorithm is cheaper than a gradient-based solution. But gradient-based solutions could be used if the attack needs to scale to substantially larger $k$.

Table 3: Performance comparisons of LLMs before and after adversarial attack. Numbers in each represent original accuracy, accuracy after adversarial attack, and relative performance drop. Red shading indicates experiments where the permutation attack reduced performance below chance level. All models suffer substantially with most experiments leading to below chance performance.

| Method | MMLU | ARC-c | BoolQ | SocialiQA | MedMCQA |
|---|---|---|---|---|---|
| Llama2-7B | 40.91/ 6.17 (34.74 ↓) | 47.04/ 7.98 (39.06 ↓) | 61.79/ 8.23 (53.56 ↓) | 52.00/15.71 (36.29 ↓) | 37.96/ 1.60 (36.36 ↓) |
| Llama2-13B | 52.22/18.33 (33.89 ↓) | 61.80/21.63 (40.17 ↓) | 67.16/38.29 (28.87 ↓) | 61.21/34.14 (27.07 ↓) | 39.78/ 7.35 (32.43 ↓) |
| Llama2-70B | 64.68/33.16 (31.52 ↓) | 80.00/51.50 (28.50 ↓) | 76.39/56.21 (20.18 ↓) | 71.60/49.85 (21.75 ↓) | 49.61/ 7.35 (32.43 ↓) |
| Vicuna-v1.5 | 48.57/18.09 (30.48 ↓) | 58.37/23.43 (34.94 ↓) | 64.04/29.60 (34.44 ↓) | 64.99/38.33 (26.66 ↓) | 39.28/ 7.67 (31.61 ↓) |
| Vicuna-v1.5-13B | 54.68/26.27 (28.41 ↓) | 69.27/38.80 (30.47 ↓) | 68.96/42.14 (26.82 ↓) | 66.07/44.42 (21.65 ↓) | 41.80/11.90 (29.90 ↓) |
| WizardLM-13B | 48.60/15.87 (32.73 ↓) | 58.20/21.12 (37.08 ↓) | 67.49/42.11 (25.38 ↓) | 63.46/31.78 (31.68 ↓) | 34.87/ 6.32 (28.55 ↓) |
| InternLM-7B | 45.72/10.45 (35.27 ↓) | 56.14/17.34 (38.80 ↓) | 65.83/26.41 (39.42 ↓) | 59.47/30.30 (29.17 ↓) | 32.63/ 2.56 (30.07 ↓) |
| InternLM-20B | 59.14/29.52 (29.62 ↓) | 78.28/54.42 (23.86 ↓) | 85.20/82.91 ( 2.29 ↓) | 79.48/65.97 (13.51 ↓) | 43.61/13.92 (29.69 ↓) |
| Falcon-7b | 31.66/ 2.49 (29.17 ↓) | 34.74/ 0.09 (34.65 ↓) | 55.35/ 2.66 (52.69 ↓) | 36.29/ 0.55 (35.74 ↓) | 28.12/ 0.07 (28.05 ↓) |
| MPT-7B | 35.60/ 3.52 (32.08 ↓) | 37.76/ 1.06 (36.70 ↓) | 58.46/ 7.03 (51.43 ↓) | 41.61/ 2.53 (39.08 ↓) | 26.31/ 1.60 (24.71 ↓) |
| GPT-3.5-turbo | 64.81/40.39 (24.42 ↓) | 82.23/61.55 (20.68 ↓) | 87.92/81.35 ( 6.57 ↓) | 70.62/56.29 (14.33 ↓) | 52.22/32.07 (20.15 ↓) |
| Random Chance | 25.0 | 25.0 | 50.0 | 33.33 | 25.0 |

and SEED-Bench (Li et al., 2023b). We use the questions in ScienceQA that have corresponding images, the MCQA subsets of MMBench, and the image-based MCQAs in SEED-Bench.

**Evaluations**  We use accuracy as our primary metric. During testing, we prompt the model to generate the possible option symbols (e.g., A to D) and extract the probability assigned to each choice in the first position. The option with the highest probability is then selected as the model's answer for that specific question. For both LLMs and VLLMs, we use greedy decoding and set the temperature to 1.

## 2.2 RESULTS

We present the main results in Table 3 and 4 for language and vision-language models respectively.

**Language Models**  In our experiments, large language models manifested a significant susceptibility to adversarial permutations, a finding consistent across various MCQA benchmarks. Our evaluation extended beyond the typical four-option MCQA datasets to include more diverse formats like the two-option BoolQ (Clark et al., 2019) and the three-option SocialIQA (Sap et al., 2019) that are naturally more resilient to the permutations. Intriguingly, the presence of only one or two distractor options did not mitigate the model's vulnerability to permutations. For instance, Llama2-7B's accuracy on BoolQ plummeted from 61.79% to a mere 8.23%, a performance even worse than random chance. Moreover, out of 50 experiments conducted with large language models, only 12 non-GPT-3.5-turbo models managed to perform better than random chance. And all of them, including GPT-3.5-turbo, suffer from significant performance decreases.

**Vision-Language Models**  In the vision-language model evaluations, the susceptibility to adversarial permutations is also severe. Despite the presence of visual context, which may intuitively add a layer of resilience, the VLLMs were not spared from the adverse effects of our permutation attacks. Among 40 experiments, 60% of the models fell below random chance performance after the adversarial attack. While InstructBLIP (Dai et al., 2023) shows relatively strong robustness to the adversarial attack. all of the models experienced significant accuracy drops ranging from 20% to 45%.

**Further Observations**  We note that within the same model family but with varying parameter sizes (e.g., InstructBLIP-7B v.s. InstructBLIP-13B), scaling up generally enhances both the baseline performance and resilience to adversarial attacks with relatively smaller declines in accuracy. We can also observe that models have different success rates over random chance in different datasets. For example, all of the LLMs failed the adversarial attack on MedMCQA dataset except GPT-3.5-turbo, which is also only slightly above the random chance. It shows the challenges of LLMs to generalize to out-of-domain data, and suggests caution about their use in unconstrained practical scenarios.

Table 4: Performance comparisons of VLLMs before and after adversarial attack. Numbers in each cell represent original accuracy, accuracy after adversarial attack, and relative performance drop. Red shading indicates performance below chance level after the permutation attack. All models suffer substantially with most experiments leading to below chance performance.

| Method | ScienceQA | A-OKVQA | SEED-Bench | MMBench |
|---|---|---|---|---|
| InstructBLIP-7B | 59.46/33.31 (26.15 ↓) | 74.06/51.62 (22.44 ↓) | 51.61/25.68 (25.93 ↓) | 64.91/41.01 (23.90 ↓) |
| InstructBLIP-13B | 64.15/41.84 (22.31 ↓) | 77.90/55.38 (22.52 ↓) | 53.65/28.79 (24.86 ↓) | 67.12/45.49 (21.63 ↓) |
| OpenFlamingo | 39.43/1.37 (38.06 ↓) | 46.90/3.58 (43.32 ↓) | 37.99/0.87 (37.12 ↓) | 38.99/5.18 (33.81 ↓) |
| Otter-Llama7B | 59.92/32.54 (27.38 ↓) | 57.99/28.30 (29.69 ↓) | 40.77/9.91 (30.86 ↓) | 55.24/19.67 (35.57 ↓) |
| Otter-MPT7B | 63.11/31.38 (31.73 ↓) | 68.21/43.19 (25.02 ↓) | 46.76/10.82 (35.94 ↓) | 61.31/36.46 (24.85 ↓) |
| LLaVA-7B | 45.20/2.28 (42.92 ↓) | 52.91/ 0.09 (52.82 ↓) | 38.36/5.67 (43.03 ↓) | 46.03/5.07 (40.96 ↓) |
| LLaVA-13B | 60.63/46.53 (14.10 ↓) | 63.14/25.85 (37.29 ↓) | 44.00/13.68 (30.32 ↓) | 59.13/31.30 (27.83 ↓) |
| Limber | 49.33/14.03 (35.30 ↓) | 39.57/1.22 (38.35 ↓) | 31.50/0.26 (31.24 ↓) | 34.93/1.62 (33.31 ↓) |
| mPLUG-Owl-pt | 53.24/10.20 (43.04 ↓) | 39.91/1.83 (38.08 ↓) | 35.57/0.91 (34.66 ↓) | 42.57/8.54 (34.03 ↓) |
| mPLUG-Owl-instr | 54.87/11.43 (43.44 ↓) | 37.12/2.01 (35.11 ↓) | 36.74/2.72 (34.02 ↓) | 43.74/6.12 (37.62 ↓) |
| Random Chance | Min 20.0 | 25.0 | 25.0 | 25.0 |

Table 5: Performance of LLMs on the MMLU dataset under answer set pruning. Numbers in each cell represent original accuracy, accuracy after adversarial attack, and relative performance drop. Baseline performances improve as the number of distractors is reduced, but performance is reduced below chance after adversarial permutation.

| Method | 4 Choices | 3 Choices | 2 Choices |
|---|---|---|---|
| Llama2-7B | 40.91 | 48.75/ 8.67 (39.08↓) | 63.33/20.26 (43.07↓) |
| Llama2-13B | 52.22 | 70.77/22.85 (47.92↓) | 71.13/31.85 (39.28↓) |
| Llama2-70B | 64.68 | 69.90/35.34 (34.56↓) | 75.23/45.88 (29.35↓) |
| Vicuna-v1.5-7B | 48.57 | 56.65/30.60 (26.97↓) | 68.81/32.60 (36.21↓) |
| Vicuna-v1.5-13B | 54.68 | 61.75/29.02 (32.66↓) | 72.97/28.06 (44.91↓) |
| WizardLM-13B | 48.60 | 56.57/17.74 (38.83↓) | 69.09/28.96 (40.13↓) |
| InternLM-7B | 45.72 | 51.76/12.39 (39.37↓) | 65.88/19.65 (46.23↓) |
| InternLM-20B | 59.14 | 65.25/30.48 (34.67↓) | 76.09/43.51 (32.58↓) |
| Falcon-7b | 31.66 | 52.88/ 5.92 (46.96↓) | 58.31/11.41 (46.90↓) |
| MPT-7B | 35.60 | 53.31/ 6.27 (47.03↓) | 58.31/15.44 (42.87↓) |
| GPT-3.5-turbo | 64.81 | 70.80/42.99 (27.81↓) | 79.30/50.82 (28.48↓) |
| Random Chance | 25.0 | 33.33 | 50.0 |

# 3 ANSWER SET PRUNING

In this section, we examine the impact of a stricter test condition on MCQA, specifically by reducing the number of distractor options, while obviously retaining the true answer. This is expected to improve baseline performance by increasing random chance level, but also we expected it to reduce vulnerability to adversarial permutation by substantially reducing the degrees of freedom that the permutation attack can explore. However, we found that models remain highly susceptible to even the few permutations available in the reduced set of options.

**Experiment Setup** Specifically, we constrain the answer set by reducing the number of total choices from four to either three or two, inclusive of the ground-truth answer. We then compare the performance metrics between these pruned sets in both permuted and non-permuted conditions to assess the relative susceptibility of the models.

**Results** We present the results of answer set pruning of MMLU datasets in Table 5 and other datasets in the appendix. As can be seen from Table 5, reducing the number of options increases the base prediction accuracy as expected, but performing adversarial permutation on the reduced answer set still dramatically reduces the accuracy even in the 2-option cases. In most cases, the performance is below the chance level given the number of options. This means that, surprisingly, even in the simplest case of a binary choice, models are not robust to whether the true answer is presented as the first or second option.

## 4 FURTHER ANALYSIS

In this section, we delve into a detailed analysis of the potential causes behind the demonstrated vulnerability, examine related attack types, explore strategies to enhance robustness, and provide qualitative examples. We refer readers to the appendix for an exhaustive set of results.

### 4.1 POSISITION BIAS AND OTHER ATTACKS

A concurrent study to ours argued for the existence of *position bias* in language model MCQA (Zheng et al., 2023a). For example, in an A/B/C/D MCQ situation, a given model might have a predisposition to selecting a particular option such as "C" and an aversion to selecting some other option such as "A", irrespective of the correctness of the answer associated with each label. Position bias could potentially explain adversarial permutation vulnerability if a model is so averse to selecting a particular option, that rotating the true answer into that slot would reliably cause it to fail.

To analyse whether position bias can explain our results, we compare our adversarial permutation results to the performance of each LLM under position bias analysis – always rotating the correct answer to a specific slot (A/B/C/D) in the answer list.

From the results in Table 6, we do see the position bias effect remarked upon by Zheng et al. (2023a). The models tested exhibit varying degrees of position bias, as results fluctuate with respect to original performance (left column). For example, Vicuna suffers limited position bias, while Falcon-7B is highly position biased. Falcon-7B's baseline accuracy of 31% rises to 70.9% when the true answer is placed in slot A – indicating a strong preference for choosing A; but drops to 3.7% when the true answer is placed in slot B, indicating a strong aversion to selecting B.

Comparing the observed position bias to the impact of our adversarial permutation, we can see that our adversarial permutation has a much stronger effect. The results after permutation (right column) are substantially worse than the position bias results. For example, Llama2-7B performs above chance level for answers in every possible position (A/B/C/D), but is reduced to below chance by our adversarial permutation. Thus we conclude that *the impact of our adversarial permutation is not explainable by position bias*. Evidently, models rely on the relationships between choices, including the distractors, which the adversarial permutation manipulates to fool them. I.e., it is not just the true answer, and the location of the true answer (position bias), but also the pattern of the distractor answers around the true answer (as explored by adversarial permutations) that determine model success or failure. This reveals a complex and concerning form of vulnerability.

Additionally, to further investigate the potential causes of the vulnerability and compare with other types of attacks, we consider circular evaluation (CircularEval) (Liu et al., 2023c) and symbol attack. Specifically, CircularEval involves rotating options while maintaining their relative positions. Symbol attack refers to using different option symbols (here we consider A/B/C/D, a/b/c/d, I/II/III/IV). In both cases, the predictions are counted as correct only if the model predicts all of the variations correctly. As shown in Table 6, while these attacks degrade performance to some extent, our adversarial attack exhibits the most substantial impact and causes the largest performance drop.

### 4.2 POST-HOC STRATEGIES FOR MITIGATION

The previous analysis of adversarial permutation vulnerability should be concerning to stakeholders interested in trustworthy and reliable AI, and suggests a new focus for researchers in developing models with improved intrinsic permutation robustness. Nevertheless, one might ask whether any post-hoc engineering fixes could alleviate this issue in practice for existing models. To this end, we explore three post-hoc strategies that have previously proven effective in improving model perfor-

Table 6: Comparison of positional bias, circular evaluation, symbol attack, and our adversarial permutation on MMLU dataset. Position bias and the other attacks have moderate impact. In contrast, our adversarial permutation severely degrades performance, usually below random chance level.

| Method | Original | A | B | C | D | CircularEval | Symbol Attack | Permutation Attack |
|---|---|---|---|---|---|---|---|---|
| Llama2-7B | 40.91 | 60.02 | 37.28 | 30.69 | 35.43 | 27.26 | 25.70 | 6.17 |
| Llama2-13B | 52.22 | 36.15 | 58.69 | 59.08 | 54.91 | 35.80 | 30.76 | 18.33 |
| Llama2-70B | 64.68 | 63.63 | 64.28 | 67.45 | 62.43 | 48.18 | 47.40 | 33.16 |
| Vicuna-7B | 48.57 | 49.83 | 63.22 | 45.46 | 37.85 | 20.23 | 33.85 | 18.09 |
| Vicuna-13B | 54.68 | 47.33 | 70.00 | 51.73 | 52.04 | 41.42 | 45.40 | 26.27 |
| WizardLM-13B | 48.60 | 34.75 | 56.38 | 45.86 | 57.56 | 22.42 | 29.07 | 15.87 |
| InternLM-7B | 45.72 | 37.23 | 65.12 | 41.49 | 42.33 | 25.23 | 29.38 | 10.45 |
| InternLM-20B | 59.14 | 51.05 | 68.75 | 53.47 | 62.35 | 34.99 | 47.06 | 29.52 |
| Falcon-7B | 31.66 | 70.86 | 3.77 | 10.52 | 14.85 | 7.69 | 14.38 | 2.49 |
| MPT-7B | 35.60 | 0.82 | 75.35 | 34.72 | 2.03 | 2.44 | 21.62 | 3.52 |
| GPT-3.5-turbo | 64.81 | 65.84 | 67.77 | 73.81 | 56.55 | 58.21 | 63.99 | 40.39 |

mance, namely, majority voting (Wang et al., 2023), contextual calibration (Zhao et al., 2021) and confidence-based voting, and ask whether they can alleviate adversarial permutation vulnerability.

**Setup** Majority voting (Wang et al., 2023) has been shown highly successful in self-ensembling over stochastic predictions. In our context, we apply it by obtaining the predictions for all possible permutations and then selecting the most frequent prediction. If most permutations lead to a correct prediction and there are only one or two pathological permutations that lead to an incorrect prediction, then majority voting should provide complete robustness to adversarial permutation. Contextual calibration (Zhao et al., 2021) is designed to mitigate the prior bias introduced from the in-context examples by estimating the model's bias toward each answer with a "content-free" query and fitting a calibration parameter. Here we consider the input question and options as the language prior bias. We first feed the model with content-free options (e.g. "N/A") as the content-free input, and then calibrate the real prediction based on the calibration parameters calculated from the content-free input. Additionally, we also apply confidence voting by taking the output that has maximum confidence among all permutations as the final prediction (M-confidence).

**Results** From the results in Table 7 for LLMs, we can see that neither defense proved effective at restoring the original performance levels. The majority voting and M-confidence certainly ameliorated the permutation attack as expected, but still fell short of the baseline accuracy with only very few models gaining improvement. This is despite their being highly impractical defenses due to imposing a $k!$-fold increase in inference cost. Contextual calibration, on the other hand, completely failed to make a meaningful impact on mitigating the adversarial attack. This re-confirms that the position bias is not the primary reason for models' permutation vulnerability.

## 4.3 ANALYSIS ON PERMUTATION DISTRIBUTION

While our main focus has been on the permutation-robustness of LLMs and VLMs, we can also ask about the distribution of responses as a function of permutation. For example, is there only one specific pathological permutation among all $k!$ options, or are there many mistake-inducing permutations? To analyse this we report in Figure 3, a histogram over the questions in ARC-challenge where each bin represents the number of questions where the specified proportion of permutations led to the correct answer that are originally correctly answered. For example, we see that Llama2-70B has a large number of questions that succeed for almost all permutations, while several models have a substantial batch of questions that are only correctly answered for around 30% of the potential permutations. Interestingly, most models have a substantial minority of questions that are only correctly answered for a small fraction of the permutations (leftmost bin).

Table 7: Impact of majority vote, contextual calibration (C-Calibration), and maximum confidence (M-Confidence) defenses against the permutation attack on the MMLU dataset. Contextual calibration fails completely. Majority vote and M-Confidence ameliorate the attack, but do not completely restore performance. Red shading indicates below-chance results.

| Method | Original | Permutation Attack | Majority Vote | C-Calibration | M-Confidence |
|---|---|---|---|---|---|
| Llama2-7B | 40.91 | 6.17 (34.74 ↓) | 33.64 (7.27 ↓) | 5.24 (35.67 ↓) | 22.62 (18.29 ↓) |
| Llama2-13B | 52.22 | 18.33 (33.89 ↓) | 48.53 (3.69 ↓) | 20.02 (32.20 ↓) | 50.83 (1.39 ↓) |
| Llama2-70B | 64.68 | 33.16 (31.52 ↓) | 65.37 (0.69 ↑) | 35.77 (28.91 ↓) | 64.20 (0.48 ↓) |
| Vicuna-v1.5-7B | 48.57 | 18.09 (30.48 ↓) | 44.10 (4.47 ↓) | 11.33 (37.24 ↓) | 38.29 (10.28 ↓) |
| Vicuna-v1.5-13B | 54.68 | 26.27 (28.41 ↓) | 52.03 (2.65 ↓) | 18.10 (36.58 ↓) | 55.58 (0.90 ↑) |
| WizardLM-13B | 48.60 | 15.87 (32.73 ↓) | 30.17 (18.43 ↓) | 8.23 (40.37 ↓) | 37.81 (11.21 ↓) |
| InternLM-20B | 59.14 | 29.52 (29.62 ↓) | 60.33 (1.19 ↑) | 28.94 (30.20 ↓) | 64.80 (5.66 ↑) |
| Falcon-7b | 31.66 | 2.49 (29.17 ↓) | 4.38 (27.28 ↓) | 3.59 (28.07 ↓) | 21.10 (10.56 ↓) |
| MPT-7B | 35.60 | 3.52 (32.08 ↓) | 13.80 (21.80 ↓) | 6.24 (29.36 ↓) | 21.42 (14.18 ↓) |

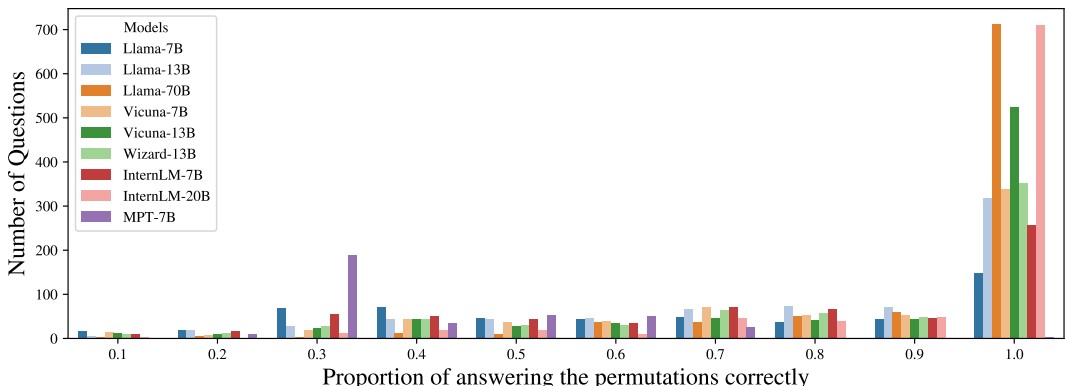

Figure 3: Analysis on permutation distribution. The histogram shows the number of questions for which the corresponding proportion of permutations leads to the correct answer (ideal is a full bar at the 100% bin, indicating that all permutations are correctly answered for all questions). The distribution of bins suggests that many questions have multiple adversarial permutations.

## 4.4 QUALITATIVE RESULTS

To illustrate the permutation attack, we present qualitative results for LLMs in Table 8 and VLLMs in Appendix Figure 5.

**Language Models** In Table 8, we showcase an MCQA example from the ARC-challenge dataset (Clark et al., 2018), with the original answer order alongside two permutations. The ground-truth answer is underlined in each configuration. We use Llama-13B for this experiment. The model gives the correct prediction for the original option order. For permutation 1, if we only swap the position of option C and D, i.e., moving the ground-truth position to C, the model can still successfully give the prediction. However, for permutation 2, even if we do not move the ground-truth answer but only swap option A and B, the model incorrectly predicts A as the answer. This qualitative example underscores that the model's vulnerability extends beyond mere positional bias and even minor changes in option ordering can result in completely different predictions.

**Vision-Language Models** In Appendix Figure 5, we present a visual MCQA example from ScienceQA dataset using Otter-Llama model. In this example, we simply move the ground truth "Asia" from option A to option C. However, the model still predicts the answer to be A and shows strong confidence in terms of the token probabilities (right part of the figure). This might show the model's preference for the first option as a recency bias.

Table 8: Qualitative results of permutations of answer options and the corresponding model (Llama2-7B) predictions. The example is selected from the ARC-challenge dataset.

| |
|---|
| **Question:** A physicist wants to determine the speed a car must reach to jump over a ramp. The physicist conducts three trials. In trials two and three, the speed of the car is increased by 20 miles per hour. What is the physicist investigating when he changes the speed? **True Answer:** the independent (manipulated) variable. |
| **Original Answer Set:** A. the control B. the hypothesis statement C. the dependent (responding) variable D. the independent (manipulated) variable. Model Prediction: D. |
| **Permutation 1:** A. the control B. the hypothesis statement C. the independent (manipulated) variable D. the dependent (responding) variable Model Prediction: C. |
| **Permutation 2:** A. the hypothesis statement B. the control C. the dependent (responding) variable D. the independent (manipulated) variable. Model Prediction: A. |

## 5 RELATED WORK

**Large Language Models and Vision-Language Models.** In recent years, the natural language processing community has seen astonishing progress in large language models (LLMs) with billions of trained parameters, such as GPT-3 (Brown et al., 2020) and Llama (Touvron et al., 2023a;b), and become more intelligent after instruction-following fine-tuning (Ouyang et al., 2022; Zheng et al., 2023b). With the strong capabilities of LLMs, there is a growing interest in grounding vision with LLMs to enable the models to perceive multimodal information (Yin et al., 2023; Zong et al., 2023; Li et al., 2023c), usually by utilizing pretrained language and vision encoders with trainable alignment modules to connect them. Such models have shown strong capabilities across a diverse range of language tasks including multimodal generation, question-answering, dialogue, and more.

**Multiple-Choice Question Answering (MCQA).** Multiple-Choice Question Answering (MCQA) requires selecting the correct option from a set of choices and is prevalent in numerous real-world applications, making it a key performance metric for both LLMs and VLLMs. Various benchmarks such as MMLU (Hendrycks et al., 2020), AGI-Eval (Zhong et al., 2023), MedM-CQA (Pal et al., 2022), and SocialIQA (Sap et al., 2019) have been designed to assess MCQA proficiency across different domains. Different prompting approaches approaches have been considered for MCQA with multiple-choice prompting being the currently recommended state of the art (Robinson & Wingate, 2023). On these benchmarks, LLMs and VLLMs frequently achieve, or even surpass, human-level accuracy (Anil et al., 2023; OpenAI, 2023b), suggesting a high degree of reliability and robustness. However, we cast doubt on this presumed robustness, exposing the underlying fragility of these models in MCQA scenarios.

**Robustness of LLMs and VLLMs.** Despite their impressive capabilities, concerns remain about the robustness and reliability of LLMs and VLLMs (Liu et al., 2023b). Previous studies have revealed the sensitivity of LLMs to various factors including prompt (Zhu et al., 2023), in-context examples (Liu et al., 2021; Zhao et al., 2021), irrelevant context (Shi et al., 2023), etc. Despite its significance, the robustness of MCQA has been relatively unexamined, particularly for VLLMs. Our research addresses this gap by scrutinizing a specific, yet pervasive, vulnerability to answer choice permutations in MCQA across both model types. Concurrent work (Zheng et al., 2023a) discusses position-bias in MCQA and Liu et al. (2023c) proposes circular evaluation. Our results show that adversarial permutation vulnerability is a much deeper problem than position bias.

## 6 DISCUSSION

In this paper, we present a comprehensive empirical analysis that unveils a critical but often overlooked vulnerability in both large language models (LLMs) and large vision-language models (VLLMs) in the context of multiple-choice question answering (MCQA). Despite their seemingly robust performance on established MCQA benchmarks, these models are highly susceptible to simple manipulations like option permutations. Our findings raise concerns about the widespread practice of evaluating and deploying these models based on MCQA tasks, urging caution in interpreting high benchmark scores as evidence of robust capabilities. We highlight the need for future work to develop training strategies and/or architectures that lead to intrinsic robustness to such adversarial attacks and develop parameter-efficient tuning approaches that can fine-tune or align existing pretrained LLMs and VLLMs to be invariant to permutations.

## REPRODUCIBILITY STATEMENT

We have attached the source code to reproduce the experimental results in the supplementary materials. All of the datasets we use are publicly available. All of the model weights (except GPT-3.5-Turbo) can be obtained from the HuggingFace model zoo or the original official Github repositories. GPT-3.5-Turbo can be accessed from OpenAI API. Experiments are conducted on A100-80GB GPUs.

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

# A APPENDIX

## A.1 ADDITIONAL RESULTS ON ANSWER SET PRUNING

We present the additional results of the answer set pruning of all language and vision-language datasets in Table 9 to 14.

## A.2 ADDITIONAL RESULTS ON DIFFERENT PROMPTING AND ATTACK STRATEGIES

In this subsection, we investigate the effect of different prompting techniques on the models' vulnerability to MCQA. The findings are summarized below.

- Table 15 compares the performance before and after adversarial attack with in-context learning. Although in-context learning can improve the original performance, the models still suffer substantially from the performance drop after adversarial permutations.

- Table 16 to Table 20 presents different attack strategies with in-context learning, i.e. permutation of in-context examples and searching for worst-case in-context examples. While they can decrease the performance, our adversarial attack has the biggest impact on the final performance and causes the largest performance drop.

- Table 21 compares different sampling strategies and temperatures. The performance of the other decoding strategies is even worse before and after the permutations compared to the greedy decoding we adopted. Therefore we can ensure our experiments were conducted properly and the findings can generalize to other decoding strategies.

- Table 22 to Table 24 presents the effect of in-context learning on the position bias. Our findings indicate that while the original model exhibited a preference for option B, this preference persisted even after introducing in-context examples with answers set to positions A, B, C, and D. This suggests that while in-context examples can modify the distribution across various options, they do not entirely override the inherent position bias of the model.

- Table 25 shows the results of a correlation analysis between predictions under different MCQ symbol sets. For each symbol set, we compute all the test set predictions under each permutations, and compute the correlation between the set of predictions made under each symbol set. A high correlation means that the behaviour in response to a permutation is similar for two symbol sets, and vice-versa. The results show a low correlation score between capital letters (A/B/C/D) and Roman numerals (I/II/III/IV) compared to the capital letters (A/B/C/D) and lowercase letters (a/b/c/d). In other words, the baseline accuracy and permuted accuracy are almost the same for different symbol sets, but they respond very differently to permutation. This suggests that the model may be relying on symbol-answer shortcuts (Geirhos et al., 2020; Du et al., 2023) and spurious correlations (Sagawa et al., 2020) inadvertently learned during training, indicating another potential underlying cause of our observed vulnerability. Figure 4 gives an illustration of the Llama2-13B model's prediction and correlation for different symbol sets.

## A.3 FURTHER ANALYSIS ON VISION-LANGUAGE DATASET

We present analysis on vision-language dataset A-OKVQA (Schwenk et al., 2022) in Table 26 and 27 about position bias and different strategies for mitigation. The additional analysis further ensures that our findings on LLMs can also be generalized to VLLMs.

Table 9: Results of answer Set Pruning on ARC (challenge) Dataset.

| Method | Original 4 Choices | 3 Choices | 2 Choices |
|---|---|---|---|
| Llama2-7B | 47.04 | 52.13/25.67 (26.46↓) | 69.44/27.04 (42.40↓) |
| Llama2-13B | 61.80 | 68.07/27.55 (40.52↓) | 77.08/39.57 (37.51↓) |
| Llama2-70B | 80.00 | 83.09/45.88 (37.21↓) | 84.21/66.01 (18.20↓) |
| Vicuna-v1.5-7B | 58.37 | 68.07/29.70 (38.37↓) | 78.11/42.66 (35.45↓) |
| Vicuna-v1.5-13B | 69.27 | 74.85/42.32 (32.53↓) | 83.18/56.14 (27.04↓) |
| WizardLM-13B | 58.20 | 67.38/28.07 (39.31↓) | 76.05/4.64 (71.41↓) |
| InternLM-7B | 56.14 | 61.37/17.42 (43.95↓) | 71.93/29.44 (42.49↓) |
| InternLM-20B | 78.28 | 82.06/48.58 (33.48↓) | 84.81/56.03 (28.78↓) |
| Falcon-7B | 34.74 | 31.76/0.00 (31.76↓) | 48.58/0.43 (48.15↓) |
| MPT-7B | 37.76 | 40.43/12.15 (28.28↓) | 50.47/0.09 (50.38↓) |
| Random Chance | 25.0 | 33.33 | 50.0 |

Table 10: Results of answer set pruning on SocialiQA Dataset.

| Method | Original 4 Choices | 2 Choices |
|---|---|---|
| Llama2-7B | 52.00 | 68.42/29.58 (38.84↓) |
| Llama2-13B | 61.21 | 73.64/45.91 (27.73↓) |
| Llama2-70B | 71.60 | 81.88/53.99 (27.89↓) |
| Vicuna-v1.5-7B | 64.99 | 73.29/41.56 (31.73↓) |
| Vicuna-v1.5-13B | 66.07 | 78.25/53.48 (24.77↓) |
| WizardLM-13B | 79.48 | 69.75/30.91 (38.84↓) |
| InternLM-7B | 59.47 | 76.41/53.02 (23.39↓) |
| InternLM-20B | 36.29 | 86.8/72.82 (13.98↓) |
| Falcon-7B | 41.61 | 55.83/0.85 (54.98↓) |
| MPT-7B | 70.62 | 60.08/4.52 (55.56↓) |
| Random Chance | 25.0 | 50.0 |

Table 11: Results of answer set Pruning on MedMCQ Dataset.

| Method | Original 4 Choices | 3 Choices | 2 Choices |
|---|---|---|---|
| Llama2-7B | 37.96 | 47.94/2.63 (45.31↓) | 62.29/5.68 (56.61↓) |
| Llama2-13B | 39.78 | 50.02/19.35 (30.67↓) | 10.37/37.45 (-27.08↓) |
| Llama2-70B | 49.61 | 55.33/17.86 (37.47↓) | 65.66/28.76 (36.90↓) |
| Vicuna-v1.5-7B | 39.28 | 46.27/10.55 (35.72↓) | 59.62/22.13 (37.49↓) |
| Vicuna-v1.5-13B | 41.80 | 50.33/24.72 (25.61↓) | 61.23/28.70 (32.53↓) |
| WizardLM-13B | 34.87 | 40.39/8.37 (32.02↓) | 52.46/10.79 (41.67↓) |
| InternLM-7B | 56.14 | 38.49/2.66 (35.83↓) | 52.02/8.42 (43.60↓) |
| InternLM-20B | 43.61 | 60.05/17.59 (42.46↓) | 65.08/30.61 (34.47↓) |
| Falcon-7B | 28.12 | 36.18/1.05 (35.13↓) | 51.30/5.23 (46.07↓) |
| MPT-7B | 26.31 | 38.33/3.07 (35.26↓) | 53.50/9.71 (43.79↓) |
| Random Chance | 25.0 | 33.33 | 50.0 |

Table 12: Results of answer set pruning on A-OKVQA dataset. Numbers in each cell represent original accuracy, accuracy after adversarial permutation attack, and relative performance drop.

| Method | 4 Choices | 3 Choices | 2 Choices |
|---|---|---|---|
| InstructBLIP7B | 74.06 | 79.21/45.92 (33.29↓) | 85.07/54.85 (30.22↓) |
| InstructBLIP13B | 77.90 | 81.66/48.33 (33.33↓) | 88.56/56.52 (32.04↓) |
| OpenFlamingo | 46.90 | 54.18/4.88 (49.30↓) | 66.90/5.09 (61.81↓) |
| Otter-Llama7B | 57.99 | 64.98/33.10 (31.88↓) | 75.02/39.74 (35.28↓) |
| Otter-MPT7B | 68.21 | 76.16/46.11 (30.05↓) | 81.48/51.44 (30.04↓) |
| Llava-7B | 52.91 | 42.86/9.44 (33.42↓) | 63.55/12.90 (50.65↓) |
| Llava-13B | 63.14 | 71.09/33.37 (37.72↓) | 76.24/41.22 (35.02↓) |
| Limber | 39.91 | 49.69/4.54 (45.15↓) | 65.68/18.08 (47.60↓) |
| mPLUG-Owl-pt | 39.91 | 45.59/4.95 (40.64↓) | 56.42/10.57 (45.85↓) |
| mPLUG-Owl-instr | 37.12 | 47.86/5.15 (42.71↓) | 58.92/16.77 (42.15↓) |
| Random Chance | 25.0 | 33.33 | 50.0 |

Table 13: Results of answer set pruning on SEED-Bench dataset. Numbers in each cell represent original accuracy, accuracy after adversarial permutation attack, and relative performance drop.

| Model | Original 4 Choices | 3 Choices | 2 Choices |
|---|---|---|---|
| InstructBLIP7B | 51.61 | 59.83/38.12(21.71 ↓) | 70.69/55.62(15.07 ↓) |
| InstructBLIP13B | 53.65 | 61.22/42.97(18.25 ↓) | 72.79/57.18(15.61 ↓) |
| OpenFlamingo | 37.99 | 39.64/10.25(29.39 ↓) | 55.31/28.35(26.96 ↓) |
| Otter-Llama7B | 50.77 | 49.46/13.48(35.98↓) | 63.57/18.86(44.71 ↓) |
| Otter-MPT7B | 46.76 | 54.18/17.42(36.76 ↓) | 66.43/28.35(38.08 ↓) |
| Llava-7B | 38.36 | 43.30/6.11(37.19 ↓) | 57.50/7.48(50.02 ↓) |
| Llava-13B | 44.00 | 52.51/17.25(35.26 ↓) | 63.37/25.24(38.13 ↓) |
| Limber | 31.50 | 38.76/1.13(37.63 ↓) | 55.58/0.06(55.52 ↓) |
| mPLUG-Owl-pt | 35.57 | 42.08/1.79(40.29 ↓) | 57.72/3.88(53.84 ↓) |
| mPLUG-Owl-instr | 36.74 | 44.35/2.94(41.41 ↓) | 56.61/7.53(49.08 ↓) |
| Random Chance | 25.0 | 33.33 | 50.0 |

Table 14: Results of answer set pruning on MMBench dataset. Numbers in each represent original accuracy, accuracy after adversarial attack, and relative performance drop.

| Method | Original 4 Choices | 3 Choices | 2 Choices |
|---|---|---|---|
| InstructBLIP7B | 64.91 | 72.61/45.28(27.33 ↓) | 79.62/51.15(28.47 ↓) |
| InstructBLIP13B | 67.12 | 72.79/50.42(22.37 ↓) | 81.27/57.30(23.97 ↓) |
| OpenFlamingo | 38.99 | 46.58/3.81(42.77 ↓) | 59.65/7.42(52.23 ↓) |
| Otter-Llama7B | 55.24 | 61.73/25.11(36.62 ↓) | 73.02/32.99(40.03 ↓) |
| Otter-MPT7B | 61.31 | 66.71/28.53(38.18 ↓) | 75.28/46.06(29.22 ↓) |
| Llava-7B | 46.03 | 45.37/2.14(43.23 ↓) | 59.42/3.66(55.76 ↓) |
| Llava-13B | 59.13 | 65.20/35.66(29.54 ↓) | 73.13/42.49(30.64 ↓) |
| Limber | 34.93 | 44.23/2.56(41.67 ↓) | 61.00/12.75(48.25 ↓) |
| mPLUG-Owl-pt | 42.57 | 53.17/12.24(40.93 ↓) | 56.42/16.90(39.52 ↓) |
| mPLUG-Owl-instr | 43.74 | 50.17/11.15(39.02 ↓) | 61.48/18.97(42.51 ↓) |
| Random Chance | 25.0 | 33.33 | 50.0 |

Table 15: Performance comparisons of LLMs before and after adversarial attack with in-context learning prompt. Numbers in each represent original accuracy, accuracy after adversarial attack, and relative performance drop. Red shading indicates experiments where the permutation attack reduced performance below chance level. All models suffer substantially with most experiments leading to below chance performance.

| Method | MMLU | ARC-c | BoolQ | SocialiQA | MedMCQA |
|---|---|---|---|---|---|
| Llama2-7B | 45.80/11.16 (34.64 ↓) | 47.04/ 7.98 (39.06 ↓) | 61.79/ 8.23 (53.56 ↓) | 52.00/15.71 (36.29 ↓) | 37.96/ 1.60 (36.36 ↓) |
| Llama2-13B | 55.37/21.69 (33.68↓) | 61.80/21.63 (40.17 ↓) | 67.16/38.29 (28.87 ↓) | 61.21/34.14 (27.07 ↓) | 39.78/ 7.35 (32.43 ↓) |
| Llama2-70B | 68.86/41.30 (27.56↓) | 80.00/51.50 (28.50 ↓) | 76.39/56.21 (20.18 ↓) | 71.60/49.85 (21.75 ↓) | 49.61/ 7.35 (42.26 ↓) |
| Vicuna-v1.5 | 49.89/19.61 (30.28↓) | 58.37/23.43 (34.94 ↓) | 64.04/29.60 (34.44 ↓) | 64.99/38.33 (26.66 ↓) | 39.28/ 7.67 (31.61 ↓) |
| Vicuna-v1.5-13B | 55.80/27.33 (28.47↓) | 69.27/38.80 (30.47 ↓) | 68.96/42.14 (26.82 ↓) | 66.07/44.42 (21.65 ↓) | 41.80/11.90 (29.90 ↓) |
| WizardLM-13B | 48.93/14.40 (34.53↓) | 58.20/21.12 (37.08 ↓) | 67.49/42.11 (25.38 ↓) | 63.46/31.78 (31.68 ↓) | 34.87/ 6.32 (28.55 ↓) |
| InternLM-7B | 48.36/15.90 (32.46↓) | 56.14/17.34 (38.80 ↓) | 65.83/26.41 (39.42 ↓) | 59.47/30.30 (29.17 ↓) | 32.63/ 2.56 (30.07 ↓) |
| InternLM-20B | 60.50/32.14 (28.36↓) | 78.28/54.42 (23.86 ↓) | 85.20/82.91 ( 2.29 ↓) | 79.48/65.97 (13.51 ↓) | 43.61/13.92 (29.69 ↓) |
| Falcon-7b | 26.95/ 0.00 (26.95↓) | 34.74/ 0.09 (34.65 ↓) | 55.35/ 2.66 (52.69 ↓) | 36.29/ 0.55 (35.74 ↓) | 28.12/ 0.07 (28.05 ↓) |
| MPT-7B | 38.73/ 5.21 (33.52 ↓) | 37.76/ 1.06 (36.70 ↓) | 58.46/ 7.03 (51.43 ↓) | 41.61/ 2.53 (39.08 ↓) | 26.31/ 1.60 (24.71 ↓) |
| Random Chance | 25.0 | 25.0 | 50.0 | 33.33 | 25.0 |

Table 16: Comparisons of different attacks on the MMLU Dataset. In-context learning (ICL) improves the zero-shot performance, and attacks on in-context examples can decrease the performance. However, our adversarial attack has the biggest impact on the final performance (largest drop).

| Model | Original 0-shot | ICL | ICL Permutation | ICL Search | Permutation Attack |
|---|---|---|---|---|---|
| Llama2-7B | 40.91 | 45.80 | 35.09 | 34.46 | 6.17 |
| Llama2-13B | 52.22 | 55.37 | 46.65 | 46.07 | 18.33 |
| Llama2-70B | 64.68 | 68.86 | 59.82 | 59.68 | 33.16 |
| Vicuna-v1.5 | 48.57 | 49.89 | 40.85 | 41.92 | 18.09 |
| Vicuna-v1.5-13B | 54.68 | 55.80 | 54.65 | 49.17 | 26.27 |
| WizardLM-13B | 48.60 | 48.93 | 39.98 | 48.27 | 15.87 |
| InternLM-7B | 45.72 | 48.36 | 37.35 | 38.17 | 10.45 |
| InternLM-20B | 59.14 | 60.50 | 54.94 | 54.45 | 29.52 |
| Falcon-7b | 31.66 | 26.95 | 27.18 | 26.79 | 2.49 |
| MPT-7B | 35.60 | 38.73 | 30.51 | 27.33 | 3.52 |

Table 17: Comparisons of different attacks on the ARC-Challenge Dataset. In-context learning (ICL) improves the zero-shot performance, and the attack on in-context examples can decrease the performance. However, our adversarial attack has the biggest impact on the final performance (largest performance drop).

| Model | Original 0-shot | ICL | ICL Permutation | ICL Search | Permutation Attack |
|---|---|---|---|---|---|
| Llama2-7B | 47.04 | 54.16 | 45.75 | 38.20 | 7.98 |
| Llama2-13B | 61.80 | 66.70 | 59.31 | 53.91 | 21.63 |
| Llama2-70B | 80.00 | 84.55 | 80.00 | 79.23 | 51.5 |
| Vicuna-v1.5 | 58.37 | 60.60 | 54.33 | 50.64 | 23.43 |
| Vicuna-v1.5-13B | 69.27 | 72.02 | 66.44 | 60.52 | 38.8 |
| WizardLM-13B | 58.20 | 59.74 | 49.01 | 43.26 | 21.12 |
| InternLM-7B | 56.14 | 65.06 | 55.54 | 51.59 | 17.34 |
| InternLM-20B | 78.28 | 80.52 | 76.74 | 74.33 | 54.42 |
| Falcon-7b | 34.74 | 37.98 | 28.46 | 22.15 | 0.09 |
| MPT-7B | 37.76 | 41.26 | 31.99 | 26.37 | 1.06 |

Table 18: Comparisons of different attacks on the BoolQ Dataset. In-context learning (ICL) improves the zero-shot performance, and the attack on in-context examples can decrease the performance. However, our adversarial attack has the biggest impact on the final performance (largest performance drop).

| Model | Original 0-shot | ICL | ICL Permutation | ICL Search | Permutation Attack |
|---|---|---|---|---|---|
| Llama2-7B | 61.79 | 63.85 | 51.49 | 40.09 | 8.23 |
| Llama2-13B | 67.16 | 65.84 | 54.95 | 54.27 | 38.29 |
| Llama2-70B | 76.39 | 84.62 | 66.42 | 55.29 | 56.21 |
| Vicuna-v1.5 | 64.04 | 69.51 | 61.47 | 57.71 | 29.60 |
| Vicuna-v1.5-13B | 68.96 | 80.24 | 71.90 | 68.23 | 42.14 |
| WizardLM-13B | 67.49 | 76.33 | 55.62 | 54.14 | 42.11 |
| InternLM-7B | 65.83 | 57.55 | 48.56 | 51.43 | 26.41 |
| InternLM-20B | 85.20 | 86.33 | 83.79 | 81.41 | 82.91 |
| Falcon-7b | 55.35 | 57.61 | 53.47 | 39.45 | 2.66 |
| MPT-7B | 58.46 | 58.99 | 55.15 | 44.02 | 7.03 |

Table 19: Comparisons of different attacks on the SocialIQA Dataset. In-context learning (ICL) improves the zero-shot performance, and the attack on in-context examples can decrease the performance. However, our adversarial attack has the biggest impact on the final performance (largest performance drop).

| Model | Original 0-shot | ICL | ICL Permutation | ICL Search | Permutation Attack |
|---|---|---|---|---|---|
| Llama2-7B | 52.00 | 57.63 | 46.37 | 33.52 | 15.71 |
| Llama2-13B | 61.21 | 67.14 | 55.78 | 45.75 | 34.14 |
| Llama2-70B | 71.60 | 75.64 | 66.99 | 64.53 | 49.85 |
| Vicuna-v1.5 | 64.99 | 64.38 | 56.81 | 47.80 | 38.33 |
| Vicuna-v1.5-13B | 66.07 | 68.58 | 58.96 | 51.07 | 44.42 |
| WizardLM-13B | 63.46 | 62.64 | 50.97 | 43.19 | 31.78 |
| InternLM-7B | 59.47 | 64.64 | 52.66 | 46.16 | 30.30 |
| InternLM-20B | 79.48 | 78.86 | 75.49 | 70.47 | 65.97 |
| Falcon-7b | 36.29 | 36.89 | 31.34 | 28.25 | 0.55 |
| MPT-7B | 41.61 | 42.91 | 33.87 | 20.78 | 2.53 |

Table 20: Comparisons of different attacks on the MedMCQA Dataset. In-context learning (ICL) improves the zero-shot performance, and the attack on in-context examples can decrease the performance. However, our adversarial attack has the biggest impact on the final performance (largest performance drop).

| Model | Original 0-shot | ICL | ICL Permutation | ICL Search | Permutation Attack |
|---|---|---|---|---|---|
| Llama2-7B | 37.96 | 39.67 | 32.10 | 26.57 | 1.60 |
| Llama2-13B | 39.78 | 39.74 | 33.24 | 28.46 | 7.35 |
| Llama2-70B | 49.61 | 51.78 | 41.37 | 41.76 | 7.35 |
| Vicuna-v1.5 | 39.28 | 38.32 | 30.72 | 27.96 | 7.67 |
| Vicuna-v1.5-13B | 41.80 | 43.22 | 36.72 | 30.68 | 11.90 |
| WizardLM-13B | 34.87 | 36.54 | 26.24 | 23.59 | 6.32 |
| InternLM-7B | 32.63 | 37.43 | 28.69 | 25.18 | 2.56 |
| InternLM-20B | 43.61 | 42.58 | 38.14 | 33.20 | 13.92 |
| Falcon-7b | 28.12 | 29.79 | 21.88 | 14.03 | 0.07 |
| MPT-7B | 26.31 | 32.24 | 19.64 | 17.05 | 1.60 |

Table 21: Results of using different sampling strategies and temperatures on MMLU dataset: before/after permutation.

| Model | Greedy Decoding | Temperature=0.5 | Temperature=1.5 | Top-k Sampling | Nucleus Sampling |
|---|---|---|---|---|---|
| Llama2-7B | 40.91/6.17 | 28.39/0.03 | 10.35/0.00 | 21.71/0.00 | 21.95/0.00 |
| Llama2-13B | 52.22/18.33 | 44.00/3.67 | 13.94/0.00 | 32.54/0.00 | 32.42/0.02 |
| Llama2-70B | 64.68/33.16 | 58.13/12.56 | 17.66/0.00 | 44.21/0.07 | 44.44/0.42 |
| Vicuna-v1.5 | 48.57/18.09 | 47.64/12.29 | 34.43/0.04 | 42.71/3.60 | 44.77/8.10 |
| Vicuna-v1.5-13B | 54.68/26.27 | 53.71/21.65 | 38.18/0.11 | 49.24/7.34 | 51.99/17.10 |
| WizardLM-13B | 48.60/15.87 | 47.56/12.43 | 38.11/0.57 | 44.61/5.86 | 45.83/10.30 |
| InternLM-7B | 45.72/10.45 | 0.01/0.00 | 0.53/0.00 | 0.16/0.00 | 0.07/0.00 |
| InternLM-20B | 59.14/29.52 | 33.53/3.89 | 19.81/0.00 | 30.82/0.31 | 31.91/1.24 |
| Falcon-7B | 31.66/2.49 | 0.02/0.00 | 0.46/0.00 | 0.06/0.00 | 0.01/0.00 |
| MPT-7B | 35.60/3.52 | 0.01/0.00 | 0.67/0.00 | 0.12/0.00 | 0.04/0.00 |

Table 22: Comparisons of position bias of Vicuna-13B with setting ground truth answers of in-context examples to specific positions.

| | A | B | C | D |
|---|---|---|---|---|
| Original positional bias | 47.33 | **70.00** | 51.73 | 52.04 |
| Moving ICL answers to A | 57.98 | **63.28** | 55.28 | 47.37 |
| Moving ICL answers to B | 58.13 | **61.39** | 56.22 | 49.02 |
| Moving ICL answers to C | 56.81 | **63.61** | 54.49 | 48.68 |
| Moving ICL answers to D | 58.02 | **60.64** | 54.45 | 50.99 |

Table 23: Comparisons of position bias of InternLM-7B with setting ground truth answers of in-context examples to specific positions.

|  | A | B | C | D |
|---|---|---|---|---|
| Original positional bias | 45.72 | **37.23** | 65.12 | 41.49 |
| Moving ICL answers to A | 31.55 | **74.76** | 44.70 | 43.42 |
| Moving ICL answers to B | 38.71 | **73.49** | 43.94 | 42.47 |
| Moving ICL answers to C | 31.58 | **69.05** | 49.27 | 46.47 |
| Moving ICL answers to D | 32.63 | **69.39** | 44.34 | 51.22 |

Table 24: Comparisons of position bias of InternLM-20B with setting ground truth answers of in-context examples to specific positions.

|  | A | B | C | D |
|---|---|---|---|---|
| Original positional bias | 51.05 | **68.75** | 53.47 | 62.35 |
| Moving ICL answers to A | 51.33 | **72.52** | 62.81 | 55.29 |
| Moving ICL answers to B | 49.70 | **73.07** | 64.34 | 56.31 |
| Moving ICL answers to C | 48.01 | **70.00** | 64.36 | 60.61 |
| Moving ICL answers to D | 46.99 | **67.54** | 62.20 | 65.72 |

Table 25: Comparisons of Pearson correlation scores of different symbol sets on ARC-Challenge dataset averaged over different models.

| Symbol Set | Pearson Correlation | Original Accuracy | Permuted Accuracy |
|---|---|---|---|
| Capital Letters vs. Lowercase Letters | 0.76 | 55.06 vs. 54.87 | 23.73 vs. 21.68 |
| Capital Letters vs. Roman Numerals | 0.36 | 55.06 vs. 52.49 | 23.73 vs. 19.33 |

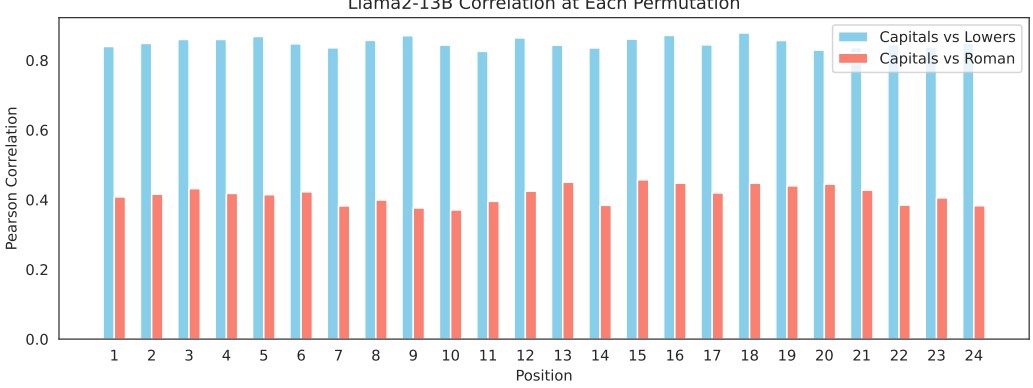

Figure 4: The correlation analysis of Llama2-13B model's predictions across different pairs of options symbols of each permutation reveals a notable finding: the low correlation score between permutation predictions when using capital letters (A/B/C/D) and Roman numerals (I/II/III/IV) suggests that the model may have learned shortcuts or spurious correlations linking option symbols with answer content.

Table 26: Comparison of positional bias and our adversarial permutation attack on A-OKVQA dataset. While position bias exists, its impact is moderate. In contrast, our adversarial method severely degrades performance, usually below random chance level.

| Method | Original | A | B | C | D | Permutation Attack |
|---|---|---|---|---|---|---|
| InstructBLIP-7B | 74.06 | 67.16 | 75.28 | 75.90 | 75.11 | 51.62 |
| InstructBLIP-13B | 77.90 | 77.29 | 72.75 | 80.35 | 73.54 | 55.38 |
| OpenFlamingo | 64.68 | 52.34 | 72.77 | 41.19 | 35.86 | 3.58 |
| Otter-Llama7B | 57.99 | 83.14 | 53.45 | 55.02 | 44.10 | 28.30 |
| Otter-MPT7B | 68.21 | 53.36 | 79.74 | 69.00 | 65.59 | 43.19 |
| LLaVA-7B | 52.91 | 77.18 | 22.71 | 14.85 | 10.94 | 0.09 |
| LLaVA-13B | 63.14 | 69.43 | 77.79 | 63.76 | 48.08 | 25.85 |
| Limber | 39.57 | 47.77 | 55.72 | 31.88 | 27.11 | 1.22 |
| mPLUG-Owl-pt | 39.91 | 33.26 | 45.16 | 47.57 | 36.49 | 1.83 |
| mPLUG-Owl-instr | 37.12 | 34.25 | 41.27 | 45.78 | 39.55 | 2.01 |

Table 27: Impact of majority vote, contextual calibration (C-Calibration), and maximum confidence (M-Confidence) defenses against the permutation attack on the A-OKVQA dataset. Contextual calibration fails completely. Majority vote and M-Confidence ameliorates the attack, but do not completely restore performance. Red shading indicates below-chance results.

| Method | Original | Adversarial Attack | Majority Vote | C-Calibration | M-Confidence |
|---|---|---|---|---|---|
| InstructBLIP-7B | 74.06 | 51.62 (22.44 ↓) | 57.47 (16.59 ↓) | 38.12 (35.94 ↓) | 69.79(4.27 ↓) |
| InstructBLIP-13B | 77.90 | 55.38 (22.52 ↓) | 60.26 (17.64 ↓) | 45.99 (31.91 ↓) | 70.83 (7.07 ↓) |
| OpenFlamingo | 46.90 | 3.58 (43.32 ↓) | 15.12 (31.78 ↓) | 7.98 (38.92 ↓) | 44.20 (2.70 ↓) |
| Otter-Llama7B | 57.99 | 28.30 (29.69 ↓) | 27.63 (30.36 ↓) | 21.33 (36.66 ↓) | 38.29 (19.70 ↓) |
| Otter-MPT7B | 68.21 | 43.19 (25.02 ↓) | 55.11 (13.10 ↓) | 42.46 (25.75 ↓) | 51.97 (16.24 ↓) |
| LLaVA-7B | 52.91 | 0.09 (52.82 ↓) | 27.86 (25.05 ↓) | 8.23 (44.68 ↓) | 50.04 (2.87 ↓) |
| LLaVA-13B | 63.14 | 29.52 (33.62 ↓) | 53.36 (9.78 ↓) | 28.94 (34.20 ↓) | 64.80 (1.66 ↑) |
| Limber | 39.57 | 1.22 (38.35 ↓) | 38.69 (0.88 ↓) | 3.59 (35.98 ↓) | 38.14 (1.43 ↓) |
| mPLUG-Owl-pt | 39.91 | 1.83 (38.08 ↓) | 14.33 (25.58 ↓) | 4.28 (35.63 ↓) | 15.21 (24.70 ↓) |
| mPLUG-Owl-instr | 37.12 | 2.01 (35.11 ↓) | 12.01 (25.11 ↓) | 2.19 (34.93 ↓) | 13.37 (23.75 ↓) |

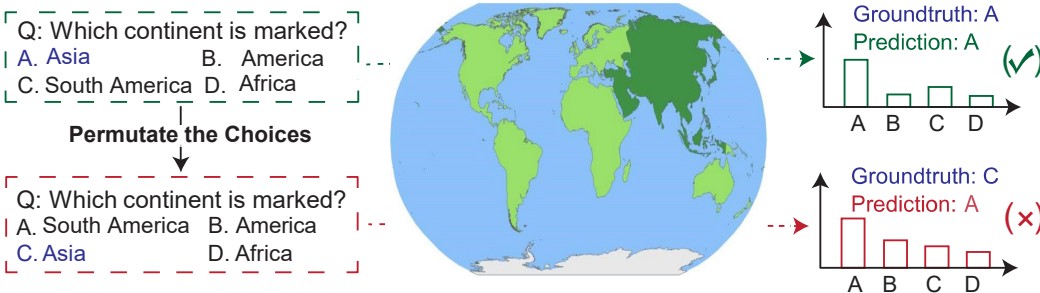

Figure 5: Qualitative results of permutations of answer options and the corresponding model (Otter-Llama) predictions. The example is selected from the ScienceQA dataset.

