# OpenReview forum: "Fool Your Large (Vision and) Language Models with Embarrassingly Simple Permutations"
_ICLR.cc/2024/Conference — Submitted to ICLR 2024_

### Official Review · Reviewer_FKE2 · 2023-10-26

**Soundness:** 3 good
**Presentation:** 3 good
**Contribution:** 3 good
**Rating:** 6
**Confidence:** 3

**Summary:**

This paper explores and experiments with the permutation attack on current large vision-language models for multiple-choice question answering (MCQA). The authors show that these models are susceptible to adversarial permutations in the answer sets for multiple-choice prompts. This is concerning because the models should be invariant to the permutations.

**Strengths:**

- Well written
- Interesting and important area of work that many are overlooking
- An important area to look at specially for industry wide adaptation of large vision-language models.
- Experiments are easy to understand

**Weaknesses:**

1. I don't see any related work section. That would have made the work more sound
2. The comparison tables on LLMs seems to be inconsistent. For example Table 3 has GPT-3.5 turbo but Table 5, 6 (that are also comparison on LLM) do not have GPT-3.5 turbo
3. On the experimentation section, it would be good if the authors described some more details. For example - did they used model APIs or model weight's for testing? This would give us an idea on the consistency of experiments

**Questions:**

1. In Table 3 for GPT3.5 turbo, how did the authors do the testing? Is it using OpenAI API?

---

> ### Author Response · Authors · 2023-11-17
> **Response**
>
> Thank you very much for reviewing our paper and providing helpful comments. Here are our responses to your comments.
>
> ### Q1. Related work section
> We would like to clarify that a comprehensive review of related work is indeed present in our paper, specifically in Section 5. This section discusses the development of Large Language Models (LLMs) and Vision-and-Language Models (VLLMs), their wide adoption, and popular studies using multiple-choice question answering for evaluation. Additionally, it includes an analysis of the robustness of LLMs and VLLMs. We hope this section can draw parallels with previous studies, and highlight the distinct contributions and originality of our work.
>
> ### Q2. Comparison of LLMs
> Thank you for pointing this out. We acknowledge that GPT-3.5 Turbo is included in Table 3 but not in Tables 5 and 6. The reason for this is primarily due to budgetary constraints. To address this issue and maintain the integrity of our comparative analysis, we have now conducted additional experiments with GPT-3.5 Turbo for Table 5 and 6. The results can be seen in the updated tables. Additionally, we also incorporated Llama2-70B, a strong open-source model, in the revised version of our manuscript. This addition aims to provide another comparable level of insight and draw similar conclusions to what would have been achieved with GPT-3.5 Turbo.
>
> ### Q3. Details of the experiments
> Thank you for your suggestions. All of the models (except GPT-3.5-turbo) we used are open-sourced models. Their weights can be obtained from the official Huggingface model zoo (https://huggingface.co/models) or the original Github repositories. We have added additional descriptions of these details in the Reproducibility Statement section and we have uploaded the source code to reproduce these experiments.
>
> ### Q4. Testing of GPT-3.5-Turbo
> Thank you for your question. Yes, we used the official OpenAI GPT-3.5-turbo API for the experiments following their documents (https://platform.openai.com/docs/api-reference/chat).

---

> > ### Comment · Reviewer_FKE2 · 2023-11-22
> >
> > Thank you to the authors for the explanation and conducting additional experiments, updating the paper.
> > I think understanding the robustness of vision-language models in general is an important line of work as more and more usecases are built around it. While the actionable steps out of this finding is not clear in this paper, I think we need to encourage this line of investigative research to the community.
> >
> > I will increase my rating to 7

---

> ### Author Response · Authors · 2023-11-21
> **Sincerely expecting further discussions from Reviewer FKE2**
>
> Dear Reviewer FKE2,
>
> We want to thank you here, again, for the constructive comments and acknowledgment of this paper. We have conducted additional experiments and provided detailed explanations to try to address all of your concerns. Could you please kindly check our revised paper and our responses, to see if your concerns are solved? We would really like to hear if you have any further questions before the discussion window is over. And if no more questions, please could you consider updating the score?
>
> Sincerely, \
> Authors

---

### Official Review · Reviewer_EvuZ · 2023-10-30

**Soundness:** 2 fair
**Presentation:** 2 fair
**Contribution:** 2 fair
**Rating:** 5
**Confidence:** 4

**Summary:**

The paper aims at the vulnerabilities of large language and vision-language models, specifically concerning permutations in multiple-choice question answering (MCQA). The authors conduct experiments that reveal performance degradation when models are exposed to permutation-based attacks, even though these models have shown capabilities in various tasks. The findings underscore the need for a deeper analysis of robustness before deploying such models in real-world applications.

**Strengths:**

The paper addresses the robustness of widely used models, a pertinent topic given the real-world deployment of these models.

**Weaknesses:**

1. The methodology's depth and novelty are not entirely clear. More analysis could be provided on how natural variations in the way are ordered may impact models.

2. While the paper focuses on permutation-based vulnerabilities, it might benefit from a broader discussion on other potential vulnerabilities or comparisons to other attack methods.

3. The experiments provided are on a specific dataset. It would be beneficial to see how the models fare on diverse datasets to ensure the findings, which are not specific to one dataset's characteristics.

**Questions:**

See the above weakness

---

> ### Author Response · Authors · 2023-11-17
> **Response (1/2)**
>
> Thank you very much for reviewing our paper and providing helpful comments. Here are our responses to your comments.
>
> ### Q1. Depth and Novelty
> *Novelty*: We would like to emphasize that the crux of our work's novelty does not lie in the complexity of the permutation method itself, but rather in the unexpected and significant findings that emerge from this straightforward yet effective approach. Our contribution is in uncovering a profound vulnerability in LLMs and VLLMs, which are otherwise regarded as highly advanced and robust. This vulnerability, triggered by simple permutations in MCQ options, reveals a critical aspect of these models that challenges their reliability in real-world applications. This phenomenon has not previously been reported in the field.
> *Depth*: The depth of our study lies in the extensive empirical analysis that such a basic manipulation can lead to drastic performance degradation. In over **100** experiments, we show that the vulnerability exists across more than 20 highly regarded foundation models and more than 9 established benchmarks. The fact that the phenomenon is so widely observed makes the result highly salient for both researchers and practitioners.
>
> ### Q2. Other attacks
> Thank you for your suggestions. In our original version, we compare our attack strategy with the position attack, i.e. always rotating the correct answers to a specific position (A/B/C/D) in the answer list. The results show that although the models also exhibit varying degrees of position bias, our adversarial permutation shows a much stronger effect (a larger proportion of predictions drop to below chance level).
> To further address your concern, we conducted additional experiments to compare with three more types of attack methods. (1) ICL Permutation attack: Given a fixed set of few-shot in-context examples, permute the in-context learning examples to their worst-case order. (2) ICL Search attack: Search for the worst-case set of ICL examples to use from within the training set.  (3) Symbol attack: Consider different types of option symbols (e.g. A/B/C/D, 1/2/3/4, a/b/c/d), and choose the worst-case symbol set for each question. We present the results of these attacks on MMLU dataset below and the results of other datasets in the appendix Table 16-20.
>
> | **Models**   	| **Original 0-shot** | **ICL** | **ICL Permutation** | **ICL Search attack** | **Symbol Attack** | **Adversarial Attack** |
> |------------------|---------------------|---------|---------------------|-----------------------|-------------------|------------------------|
> | Llama2-7B    	| 40.91           	| 45.67   | 35.09           	| 34.46             	| 25.70         	| **6.17**           	|
> | Llama2-13B   	| 52.22           	| 54.52   | 46.65           	| 46.07             	| 30.76         	| **18.33**          	|
> | Llama2-70B   	| 64.68           	| 68.25   | 59.82           	| 59.68             	| 47.40         	| **33.16**          	|
> | Vicuna-v1.5  	| 48.57           	| 50.27   | 40.85           	| 41.92             	| 33.85         	| **18.09**          	|
> | Vicuna-v1.5-13B  | 54.68           	| 55.68   | 54.65           	| 49.17             	| 45.40         	| **26.27**          	|
> | WizardLM-13B 	| 48.60           	| 48.93   | 39.98           	| 39.90             	| 29.07         	| **15.87**          	|
> | InternLM-7B  	| 45.72           	| 48.36   | 38.17           	| 38.17             	| 29.38         	| **10.45**          	|
> | InternLM-20B 	| 59.14           	| 60.50   | 54.45           	| 54.45             	| 47.06         	| **29.52**          	|
> | Falcon-7B    	| 31.66           	| 32.95   | 27.18           	| 26.79             	| 14.38         	| **2.49**           	|
> | MPT-7B       	| 35.60           	| 38.73   | 30.51           	| 27.33             	| 21.62         	| **3.52**           	|
>
>
> As can be seen from the table, while other attacks can also harm the performance, our adversarial attack has the biggest impact on the LLMs and causes the largest performance drop.

---

> > ### Author Response · Authors · 2023-11-17
> > **Response (2/2)**
> >
> > ### Q3. Evaluation datasets
> > We appreciate the reviewer's comment on the dataset specificity of our experiments. We would like to clarify that our study encompasses a broad spectrum of datasets: 5 NLP datasets for Large Language Models (LLMs) and 4 vision-language datasets for Vision-Language Large Models (VLLMs). These datasets are meticulously chosen to evaluate a diverse range of characteristics, such as commonsense reasoning, reading comprehension, external knowledge, and visual grounding abilities. The composition and specifics of these datasets are detailed in Tables 1 and 2, while the main results are presented in Tables 3 and 4. Also, in appendix Table *9-27*, we have provided the detailed results of the other datasets.
> >
> > Our analysis in Section 4, using the MMLU dataset as a case study, aims to delve deeper into the potential vulnerabilities and their remedies. MMLU is a comprehensive benchmark aggregating 57 diverse tasks, encompassing areas like mathematics, US history, computer science, and law. We believe this diversity supports the generalizability of our findings. However, to address your concern, we conducted additional analysis with another vision-language dataset A-OKVQA, and have updated the results in the appendix Table 27. The supplemental results align with our original findings, further solidifying the generalizability of our analysis.

---

> ### Author Response · Authors · 2023-11-21
> **Sincerely expecting further discussions from Reviewer EvuZ**
>
> Dear Reviewer EvuZ,
>
> We want to thank you here, again, for the constructive comments and acknowledgment of this paper. We have conducted additional experiments and provided detailed explanations to try to address all of your concerns. Could you please kindly check our revised paper and our responses, to see if your concerns are solved? We would really like to hear if you have any further questions before the discussion window is over. And if no more questions, please could you consider updating the score?
>
> Sincerely, \
> Authors

---

### Official Review · Reviewer_RBKq · 2023-10-31

**Soundness:** 2 fair
**Presentation:** 3 good
**Contribution:** 3 good
**Rating:** 6
**Confidence:** 4

**Summary:**

This paper examines permutation sensitivity in MCQA for generative language models and vision-language models. They show that a wide range of models display severe permutation sensitivity in a variety of MCQA benchmarks, and show that existing mitigation strategies do not solve permutation sensitivity.

**Strengths:**

A large number of language models and vision language models are used in the experiments. The datasets chosen are diverse and there are enough datasets to draw strong conclusions.

There is concurrent work in the area (pointed out by the authors themselves), but the broadness of the experimental evaluation is original. While limited in scope to MCQA, it is significant, as it may point to underlying problems in LLM reasoning that cannot be easily fixed.

**Weaknesses:**

I think it's overclaiming to call this an "adversarial" attack. Permuting the choices is done as a matter of course in evaluation (see Section 4.1 in [1]).

Additionally, I think important context is missing. We know parameters like the temperature and the sampling strategy have a significant effect on output. But I don't see these numbers reported. Do the numbers change if we reduce / increase the temperature?

There's no discussion on prompting. Does providing in-context examples effect permutation sensitivity? In-context examples for MCQA should always be available. They also provide a convenient way to alter the posterior distribution (for example, you could set the answer to always be the last answer in the in-context examples; would the positional bias then change?)

Also, no information is reported on the model perplexity / confidence during permutation. It may be possible that certain permutations have much lower perplexity / higher confidence, and hence the model's answer on those is more trustworthy. So in practice, we might evaluate the model on all the permutations, then select the model's most confident answer. Of course, if we can do this, it is better to avoid MCQA entirely and have the model assess the answers one-by-one.

[1] MMBench: Is Your Multi-modal Model an All-around Player?

**Questions:**

Please see the weaknesses section. I am willing to raise my rating if the weaknesses are addressed.

---

> ### Author Response · Authors · 2023-11-17
> **Response (1/2)**
>
> Thank you very much for the thoughtful review and excellent questions. Here are our responses to your comments.
>
> ### Q1. Adversarial attack v.s. Permutation
> The term “adversarial attack” in the context of machine learning generally connotes any deliberate attempt to deceive or manipulate a model's output. Adversarial examples refer to an input that is crafted to look normal to humans while triggering erroneous behaviour from the model. We argue that MCQ permutation falls within this scope, as humans do not care about the permutation of the candidate choices, but we show that a worst-case permutation dramatically worsens model performance in a way that is not anticipated or intended by the model's design.
>
> Secondly, we would like to emphasize that our choice of terminology was not meant to suggest a sophisticated or elaborate attack mechanism. On the contrary, the core message of our paper is to highlight the surprising fact that even the most straightforward forms of manipulation, such as the proposed permutation, can significantly impair the performance of advanced LLMs and VLLMs in MCQ scenarios. The fact that such a simple attack is successful only enhances the salience of our result. This finding is crucial as it underscores the need for future work to improve the robustness of these models against even the simplest forms of potential manipulation.
>
> Lastly, we are grateful for your suggestion of MMBench. We emphasize that MMbench performs *circular rotation* of answer candidates, not *permutation* of answer candidates. I.E: For N candidate answers, MMBench tries $N$ orders, preserving the relative order of the options, while we consider all possible $N!$ permutations, breaking the relative order of the options. In the revised version, we have added extra experiments comparing our permutation approach with MMBench circular evaluation (updated Table 6). The results show that our permutation results in much worse performance, demonstrating a stronger attack. The distinct outcomes of rotation and permutation are a critical step toward understanding this vulnerability and ultimately defending against it.
>
> ### Q2. Temperature and sampling strategies
> Thank you for your question. For all of the experiments, we consistently set the temperature to 1 and used greedy decoding for the generation to ensure reproducibility (specified in section 2.1 “Evaluation” paragraph). Specifically, we follow the common practice for MCQ evaluation [1][2] to obtain the probabilities of the first token and select the one with the highest probability, comparing it against the ground truth answer. (Note that in the MCQ context, we only need to select one token to answer.) Under this greedy evaluation scheme, variations in temperature or the adoption of different sampling strategies do not impact the results. This actually further enhances the generalizability and importance of our results for MCQ evaluation.
>
> However, to address your concern, we performed additional experiments using different temperatures (0.5, 1.5) and two commonly used sampling strategies (top k sampling and nucleus Sampling). As can be seen from the table below, the greedy decoding strategy we adopted achieves the best accuracy before and after the permutation among different decoding strategies. And the performance of the other decoding strategies are even worse after the permutations. This confirms both that our initial experiments were optimally configured, and also that the findings generalize to other decoding strategies. This analysis has been added to Appendix A.2 of the paper.
>
> | **Model**         	| **Greedy Decoding** 	| **Temperature=0.5**  | **Temperature=1.5**  | **Top-k Sampling**  | **Nucleus Sampling** |
> |-----------------------|----------------|------------------|------------------|-----------------|------------------|
> | Llama2-7B     	| **40.91/6.17** | 28.39/0.03   	| 10.35/0.00   	| 21.71/0.00  	| 21.95/0.00   	|
> | Llama2-13B   	| **52.22/18.33**| 44.00/3.67   	| 13.94/0.00   	| 32.54/0.00  	| 32.42/0.02   	|
> | Llama2-70B    | **64.68/33.16**| 58.13/12.56  	| 17.66/0.00   	| 44.21/0.07  	| 44.44/0.42   	|
> | Vicuna-v1.5    | **48.57/18.09**| 47.64/12.29  | 34.43/0.04   	| 42.71/3.60  	| 44.77/8.10   	|
> | Vicuna-v1.5-13B  | **54.68/26.27**| 53.71/21.65  	| 38.18/0.11   	| 49.24/7.34  	| 51.99/17.10  	|
> | WizardLM-13B     | **48.60/15.87**| 47.56/12.43  	| 38.11/0.57   	| 44.61/5.86  	| 45.83/10.30  	|
> | InternLM-7B   | **45.72/10.45**| 0.01/0.00   | 0.53/0.00    	| 0.16/0.00   	| 0.07/0.00    	|
> | InternLM-20B     	| **59.14/29.52**| 33.53/3.89   	| 19.81/0.00   	| 30.82/0.31  	| 31.91/1.24   	|
> | Falcon-7B         | **31.66/2.49** | 0.02/0.00    	| 0.46/0.00    	| 0.06/0.00   	| 0.01/0.00    	|
> | MPT-7B            	| **35.60/3.52** | 0.01/0.00    	| 0.67/0.00    	| 0.12/0.00   	| 0.04/0.00    	|
>
> [1] Measuring Massive Multitask Language Understanding. Hendrycks et al., ICLR 2021.\
> [2] Holistic evaluation of language models. Liang et al., 2022.

---

> > ### Author Response · Authors · 2023-11-17
> > **Response (2/2)**
> >
> > ### Q3. Different prompting
> > Thank you for bringing up this interesting point. Our initial experiments focused on zero-shot MCQA. However, we agree that in-context examples could have an influence on permutation sensitivity in a few-shot context. To address your concern, we conducted two categories of experiments to demonstrate the effect of in-context examples for all of the LLMs. We didn’t conduct these experiments with VLLMs because most of them do not support in-context learning. We summarize the findings below and report the raw numbers in Appendix A.2, Table *15-24*.
> >
> > (1) Re-evaluation of our zero-shot MCQ analysis under in-context learning conditions.
> > In this experiment, we randomly sample 4-shot in-context examples from the training or validation set (whichever is available) and compare the performance before and after permutation of the query options. While the original performance can be improved by in-context learning, the LLMs are still vulnerable to the permutation of option orders. As can be seen from Appendix Table 15, **74%** of the total experiments drop to below chance level even with the in-context examples and all of them suffer from huge performance drops. *In summary: In-context learning does not alleviate the issue of vulnerability to permutation attack.*
> >
> > (2) The performance after moving all of the ground truth answers of in-context examples to certain positions.
> > In this experiment, we randomly sample 4-shot in-context examples, and move all of the ground truth answers to a certain option position, repeating this process for each position (A/B/C/D) and recording the performance. We then compared the position bias of the original prompt with the bias observed after providing in-context examples that positioned answers in predetermined slots. The results for the InternLM-20B model are presented below, with detailed results for other models included in the appendix. Our findings indicate that while the original model exhibited a position-bias preference for option B, this preference persisted even after introducing in-context examples with answers set to positions A, B, C, and D. This suggests that while in-context examples can modify the position-bias across various options, their impact is limited. *In summary: Models broadly maintain their zero-shot position bias despite attempts to override it by introducing conflicting position bias to the ICL training set. *This suggests that the effects that we study are dominated by the pre-trained model and not strongly influenced either way by ICL.*
> >
> > |   	MMLU / InternLM-20B        	| A 	| B   | C 	| D 	|
> > |---------------------------|-------|---------|-------|-------|
> > | Original positional bias  | 51.05 | **68.75** | 53.47 | 62.35 |
> > | All ICL answers set to A   | 51.33 | **72.52** | 62.81 | 55.29 |
> > | All ICL answers set to B   | 49.70 | **73.07** | 64.34 | 56.31 |
> > | All ICL answers set to C   | 48.01 | **70.00** | 64.36 | 60.61 |
> > | All ICL answers set to D   | 46.99 | **67.54** | 62.20 | 65.72 |
> >
> > ### Q4. Model perplexity and confidence
> >
> > Thanks for your suggestion. To address your concern, we have conducted experiments by taking the most confident answer as the final answer among all permutations. The results are shown below (the M-confidence column) and updated in Table 7.
> >
> >
> > | **Method**  	 | **Original** | **Majority Vote**  	 | **M-Confidence**  		 |
> > |------------------|----------|-----------------------------|----------------------------|
> > | Llama2-7B   	 | 40.91    | 33.64 (7.27 $\downarrow$)   | 22.62 (18.29 $\downarrow$) |
> > | Llama2-13B  	 | 52.22    | 48.53 (3.69 $\downarrow$)   | 50.83 (1.39 $\downarrow$)  |
> > | Llama2-70B  	 | 64.68    | 65.37 (0.69 $\uparrow$)     | 64.20 (0.48 $\downarrow$)  |
> > | Vicuna-v1.5-7B   | 48.57    | 44.10 (4.47 $\downarrow$)   | 38.29 (10.28 $\downarrow$) |
> > | Vicuna-v1.5-13B  | 54.68    | 52.03 (2.65 $\downarrow$)   | 55.58 (0.90 $\uparrow$)    |
> > | WizardLM-13B     | 48.60    | 30.17 (18.43 $\downarrow$)  | 37.81 (11.21 $\downarrow$) |
> > | InternLM-20B     | 59.14    | 60.33 (1.19 $\uparrow$)     | 64.80 (5.66 $\uparrow$)    |
> > | Falcon-7b   	 | 31.66    | 4.38 (27.28 $\downarrow$)   | 21.10 (10.56 $\downarrow$) |
> > | MPT-7B  		 | 35.60    | 13.80 (21.80 $\downarrow$)  | 21.42 (14.18 $\downarrow$) |
> >
> > It can be seen that M-confidence can ameliorate the permutation attack to some extent – similarly to the impact of the majority vote defence. However, it is still generally far below the baseline un-permuted performance, indicating that the problem still exists. And the $O(n!)$ time complexity of this strategy, as you pointed out, prevents it from being a practical way to mitigate the vulnerability.

---

> ### Author Response · Authors · 2023-11-21
> **Sincerely expecting further discussions from Reviewer RBKq**
>
> Dear Reviewer RBKq,
>
> We want to thank you here, again, for the constructive comments and acknowledgment of this paper. We have conducted additional experiments and provided detailed explanations to try to address all of your concerns. Could you please kindly check our revised paper and our responses, to see if your concerns are solved? We would really like to hear if you have any further questions before the discussion window is over. And if no more questions, please could you consider updating the score?
>
> Sincerely,\
> Authors

---

> ### Comment · Reviewer_RBKq · 2023-11-23
> **Response to authors**
>
> The authors have answered my questions. I have no more questions. As promised, I am increasing my score.

---

### Official Review · Reviewer_d9tg · 2023-11-01

**Soundness:** 3 good
**Presentation:** 3 good
**Contribution:** 3 good
**Rating:** 5
**Confidence:** 3

**Summary:**

This paper reveals an interesting phenomenon of existing LLM and VLLMs: they are vulnerable to adversarial permutation in answer sets for multiple-choice prompting, which is surprising as models should ideally be as invariant to prompt permutation as humans are.

**Strengths:**

1. This paper reveals an interesting phenomenon:  they are vulnerable to adversarial permutation in answer sets for multiple-choice prompting, which they should not be ideally.

2. Experiments are conducted across different multiple LLM / VLLMs, demonstraing the universality of the phenomenon.

**Weaknesses:**

While the finding is interesting, I wonder if authors could provide any intuitive explanations on the observation? Position bias (Zheng et al., 2023a) may not be able to fully explain this phenomenon, but it can be potentially one of the reasons why they are vulnerable to the adversarial permutation in answer sets from my understanding. I am willing to see more analysis on the potential causes of this observation. Can the explanations on similar problems in other tasks besides MCQA apply to this case？Authors may put more efforts on it.

**Questions:**

Please refer to Weaknesses

---

> ### Author Response · Authors · 2023-11-17
> **Response**
>
> Thank you very much for your review and questions. Here are our responses to your comments.
>
> ### Q1. Intuitive explanations of the phenomenon
> We have further delved into it and would like to present our additional findings:
> 1. **Option Symbol-Answer Content Shortcut**: Beyond position bias, we hypothesized another contributing factor: the model may suffer from shortcut [1] or spurious correlation learning [2] between option symbols and answer content. To investigate this, we conducted three experiments using different answer symbol sets (A/B/C/D, a/b/c/d, and I/II/III/IV). We examined if models exhibit similar patterns in response to the same permutations but only with different symbols, keeping the answer content constant. We conducted experiments on ARC-Challenge dataset with all models. Interestingly, our results revealed a high correlation for similar symbol pairs (A/B/C/D and a/b/c/d) with a Pearson correlation score of 0.76. However, for more distinct symbol pairs (A/B/C/D and I/II/III/IV), the correlation score dropped to 0.36, indicating that the models’ response to permutation differs significantly when only the symbols varied. In other words, the baseline accuracy and permuted accuracy are almost the same for different symbol sets, but they respond very differently to permutation. This discrepancy points to the models having learned shortcuts between symbols and answer content, which may give extra vulnerability to the permutations of the models. This offers an additional explanation for the observed phenomenon. This analysis has been added to Appendix A of the paper.
>
> | Symbol Set              	| Correlation | Original Accuracy | Permuted Accuracy |
> |-----------------------------|-------------|-------------|-------------|
> | Capital Letters vs. Lowercase Letters | 0.76  |  55.06 vs. 54.87 	|   23.73 vs. 21.68	|
> | Capital Letters vs. Roman Numerals	| 0.36  |  55.06 vs. 52.49	|   23.73 vs. 19.33 |
>
> 2. **Bias from the Training Set**: We also considered data in the training set might lead to the observed vulnerability in MCQA [3]. Training data biases can arise from the predominance of certain answer patterns or thematic content. For instance, if the training set disproportionately represents specific types of questions or contextual themes, the model may develop an over-tuned response to these patterns. This over-tuning could result in a skewed performance when the model encounters permutations in MCQs that deviate from these familiar patterns. However, given that current LLMs and VLLMs are trained on vast and varied datasets, quantifying the exact impact of these training data biases on model vulnerability remains a challenging endeavor and we leave it for future work.
>
>
> [1] Geirhos, Robert, et al. "Shortcut learning in deep neural networks." Nature Machine Intelligence 2020.\
> [2] Sagawa, Shiori, et al. "An investigation of why overparameterization exacerbates spurious correlations." ICML 2020.\
> [3] Du, Mengnan, et al. "Shortcut learning of large language models in natural language understanding: A survey." arXiv 2022.

---

> ### Author Response · Authors · 2023-11-21
> **Sincerely expecting further discussions from Reviewer d9tg**
>
> Dear Reviewer d9tg,
>
> We want to thank you here, again, for the constructive comments and acknowledgment of this paper. We have conducted additional experiments and provided detailed explanations to address your concerns. Could you please kindly check our revised paper and our responses, to see if your concerns are solved? We would really like to hear if you have any further questions before the discussion window is over. And if no more questions, please could you consider updating the score?
>
> Sincerely,\
> Authors

---

### Author Response · Authors · 2023-11-17
**General Response from Authors**

Dear AC and reviewers,

We sincerely thank AC and all reviewers’ time and efforts in reviewing our paper. The constructive suggestions have helped us to improve our paper further. We appreciate that all of the reviewers find the problem we study interesting and important, as well as the positive acknowledgment of the comprehensiveness of our experimental approach.

We have additionally conducted extensive experiments and updated the manuscript according to the reviewers' comments and suggestions (updated main text and Appendix Table 16-27). Here is a summary of our updates:
- [**Analysis of the potential cause**] Besides position bias, we investigated models relying on shortcuts between option symbols and answer content as a potential flaw that is exploited by the permutation attack. This shortcut effect is demonstrated by analysis of correlation scores across different symbol set choices.
- [**In-context Learning**] Our experiments show that in-context learning (1) improves original performance but doesn't reduce permutation vulnerability, and (2) also does not override inherent position bias.
- [**Model perplexity and confidence**] We conducted experiments using the most confident answer from permutations as a potential defense against the attack (updated in Table 7). This ameliorates the attack similarly to the majority vote defense, but still suffers a substantial performance drop overall. And similarly to the majority vote defense, it still requires an impractical $O(n!)$ time complexity.
- [**Comparison with other types of attacks**] Our revised experiments compare our adversarial permutation with three additional attack methods. The results show that while other attacks also affect performance, our permutation attack is the strongest, resulting in the most significant performance drop.
- [**Clarification of experimental details**] We have further explained the details that reviewers are interested in or feel unclear about. The corresponding contents have been added or modified in our revised version. Source code is also uploaded.

---

### Meta-Review · Area_Chair_FYUu · 2024-01-05

**Metareview:**

The present paper addresses questions related to robustness of large language and vision-language models, specifically in the context of these models' capability to answer multiple choice questions (MCQs). The paper finds that current models are specifically vulnerable to adversarial ordering of the presented answers.

Initial reviews for this paper identified a number of strengths and weaknesses. Reviewers were positive about the interesting phenomenon identified by the authors, and the well-designed experiments, and the large number of models compared. Weaknesses included that different sets of models were included across the presented experiments (due to budget constraints), missing details of experiments, the limited number of benchmark data sets, and thoughts that calling the approach an "adversarial attack" might be overclaiming. Reviewers also noted concerns about the methods' novelty, and the overall depth and significance of the presented insights.

The authors provided a rebuttal, including additional empirical analysis. There was discussion between reviewers and authors, as well as a separate reviewer-AC discussion. Reviewers acknowledged the authors' efforts during the rebuttal and discussion phase. Several concerns were addressed and reviewers revised their scores upwards. At the same time, the paper remained borderline at the end of the rebuttal period. Follow-up with reviewers found that two had substantial remaining concerns. In particular, they found that the overall breadth and depth of insight presented in the current manuscript was not sufficient for acceptance at ICLR. More positive reviewers noted that they would very much like to encourage this important line of work, but agreed that the depth of actionable insights provided was not substantial. As a result, the AC sides with the more negative reviewers and recommends that the manuscript is not ready for publication at ICLR at the present stage.

**Justification For Why Not Higher Score:**

The paper does not make a substantial enough contribution to this field.

**Justification For Why Not Lower Score:**

N/A

---

### Decision · Program_Chairs · 2024-01-16

Reject